# Low-Cost Eye Tracking Calibration: A Knowledge-Based Study [note 1]

**DOI:** 10.3390/s21155109

**Published:** 2021-07-28

**Authors:** Gonzalo Garde, Andoni Larumbe-Bergera, Benoît Bossavit, Sonia Porta, Rafael Cabeza, Arantxa Villanueva

**Affiliations:** 1Department of Electrical, Electronic and Communications Engineering, Arrosadia Campus, Public University of Navarre, 31006 Pamplona, Spain; andoni.larumbe@unavarra.es (A.L.-B.); sporta@unavarra.es (S.P.); rcabeza@unavarra.es (R.C.); avilla@unavarra.es (A.V.); 2School of Computer Science and Statistics, Trinity College Dublin, The University of Dublin, College Green, D02 PN40 Dublin 2, Ireland; bossavib@scss.tcd.ie

**Keywords:** gaze-estimation, calibration, low-resolution, theoretical analysis

## Abstract

Subject calibration has been demonstrated to improve the accuracy in high-performance eye trackers. However, the true weight of calibration in off-the-shelf eye tracking solutions is still not addressed. In this work, a theoretical framework to measure the effects of calibration in deep learning-based gaze estimation is proposed for low-resolution systems. To this end, features extracted from the synthetic U2Eyes dataset are used in a fully connected network in order to isolate the effect of specific user’s features, such as kappa angles. Then, the impact of system calibration in a real setup employing I2Head dataset images is studied. The obtained results show accuracy improvements over 50%, probing that calibration is a key process also in low-resolution gaze estimation scenarios. Furthermore, we show that after calibration accuracy values close to those obtained by high-resolution systems, in the range of 0.7°, could be theoretically obtained if a careful selection of image features was performed, demonstrating significant room for improvement for off-the-shelf eye tracking systems.

## 1. Introduction

In recent years, several areas within the world of computer vision have undergone a revolution with the irruption of deep neural networks, systems that have allowed to change the problem-solving approach by shifting the focus towards the quality of the training data used rather than towards the solving method itself. With neural networks, outstanding results have been obtained in domains such as object classification, image generation and pose estimation [1,2,3]. The field of gaze tracking has not been an exception and has been favored by the emergence of this research stream, with an increasing number of articles proposing the use of deep neural networks.

However, one of the main problems with the use of neural networks is their black box characteristic, which prevents researchers from clearly understanding what the fundamental part of the data is that is used to feed the network, i.e., how it is addressing and solving the problem. This, from a research point of view, can be negative as it hinders the establishment of a theoretical framework, which defines what elements should appear as input data to make the problem solvable. Of course, this degree of uncertainty is not the same for all types of neural networks: in the case of Convolutional Neural Networks (CNNs) where the input to the network consists of images, the uncertainty stands on the key features of the image that allow the problem to be solved. On the other hand, in the case of Fully Connected Networks (FCNs), the uncertainty is less significant since researchers decide which inputs should be introduced to the network. In this situation, a thorough prior knowledge of the problem to be solved is required to effectively select the features and the network to be used.

Providing degrees of freedom to neural networks to assess which features play crucial roles is one of the virtues of neural networks since it does not require prior knowledge and it prevents errors due to false assumptions from an incomplete knowledge about the problem model. However, from a researchers’ point of view, it becomes more difficult to understand the learning process of the networks. In the end, the best solution lies somewhere in between, combining both knowledge-based models and more blind approaches, such as neural networks.

Understanding how neural networks learn to solve a problem is a field in continuous development, with methodological proposals that help us assess and evaluate how neural networks learn. As mentioned above, as there are different flavors of neural networks, different strategies are used for each type: to give some examples, in the case of FCNs, we would find the method of individual randomization of each of the features to measure their impact on the final performance [4] or the use of saliency maps for Convolutional Neural Networks, with the aim of learning which region of the image has a higher activation when inferring a result [5].

Furthermore, by the very nature of the real databases used to train the models, it cannot be guaranteed that all the features relevant to the resolution of a problem appear in all the images or inputs. If the database is sufficiently representative of the environment of the problem, this should not be an issue and may even provide the network with greater generalization capacity, although the lack of certainty about the ground-truth values also hinders the work of developing a theoretical framework. It is here where synthetic databases, which provide absolute control of the ground-truth, can be an essential support point for the development of research in this field. In the last few years, works from different areas have benefited from training over synthetic environments [6,7,8,9].

Within the field of gaze-estimation, we can distinguish between two approaches: high-resolution and low-resolution scenarios. High-resolution comprises those systems that, using specialized hardware and special conditions such as infrared (IR) lighting or limited head movements, perform the gaze-estimation process on images in which the eye region occupies a high number of pixels, hence the name high-resolution. These specialized conditions together with the need of IR lighting needed to correct the performance of these systems make plug-and-play implementation difficult. On the other hand, low-resolution systems are those in which the aim is to make use of more general images, such as those that could be obtained from mobile devices or webcams, to perform gaze-estimation. This is attractive because it would allow gaze-estimation applications to be expanded to a broader audience. In the case of high-resolution systems, it can be considered that the problem of gaze-estimation has been solved since high-performance commercial devices are available. However, these models cannot be applied straightforwardly in low-resolution environments due to changes in image quality and illumination among others, which has led to the use of models based on neural networks for low-resolution [10,11,12].

Gaze-estimation presents the same problems as in other fields when facing deep neural networks approaches: it is difficult to determine whether the features that these systems extract from the images agree with the previous knowledge about high-resolution systems and, if so, to what extent. Moreover, we can neither determine whether the differences between high and low-resolution gaze-estimation systems are due to physical limitations, i.e., the same results cannot be achieved in high and low-resolution; to limitations during training, i.e., we have reached the training limits of neural network for this problem; nor due to a problem in the extracted features.

The need for calibration for eye-tracking, to adapt the model to the characteristics of the user, is well known due to the previous work regarding high-resolution systems. The user calibration stage allows for better accuracies. In high-resolution, this calibration is based on adjusting the coefficients of a polynomial, or on the calculation of some parameters of a model, with the kappa angles being one of the fundamental ones. As an introduction, and although it will be further developed in later sections, kappa angles represent the angular offsets between the optical axis, the symmetry axis of the eyeball, and the visual axis, that is approximated by the line between the fixation point and the projection of the image on the fovea (Line of Sight, LoS). These kappa angles are not visible in the image, so end-to-end CNN-based models are not enough to determine it alone.

Calibration therefore plays an important role in obtaining an accurate gaze-estimation. Traditionally, the calibration process consisted of presenting a series of points on a screen for the user to fixate on. From these known points and the capture of the user’s gaze, the individual’s own characteristics can be inferred. The main drawback with this procedure is that it has been reported as inorganic, difficult for some users and tedious [13]. Ultimately, the quality and reliability of the data obtained may be compromised, thus losing the advantage of calibration. To alleviate this drawback, in recent years, solutions have been proposed that seek to facilitate the calibration process, so that it becomes a simpler, more reliable and faster procedure to perform. Within this line of solutions, we find proposals such as [14,15,16] based on pursuit eye movement, which constitutes a more organic process. This is critical for the usability of future products, as pointed out by [17], and should be an aspect of concern in the development of new solutions.

The high-resolution paradigm has led to the use of calibration steps in low-resolution gaze-estimation models to improve the results obtained. The need for user calibration is more clearly and frequently spoken of in some of the most recent works, as in [18,19] or [20]. Most of the works use blind calibration methods. This tendency, in addition to providing simplicity, is partly derived from the paradigm used in high-performance eye tracking systems in which methods, for example, based on regressions on generic polynomial models, gave good results. Few exceptions, such as [21] or [12,22], propose a more geometry-based approach to the calibration/personalization problem. However, due to the existing differences between high and low-resolution gaze-estimation systems, it is necessary to try to substantiate a theoretical framework to evaluate the true weight of calibration in a low-resolution gaze-estimation approach. To the best of our knowledge, this is the first paper to make a detailed study of this topic.

The main motivation of this paper is to build a theoretical framework on the influence of calibration specific to low-resolution that allows us to advance in a differentiated way on high-resolution, alleviating dependence among the two. To build this framework, we will use a synthetic environment for its elaboration, which allows us to have absolute control over the features used, and a real scenario to test the findings.

In this paper, we will try to establish a theoretical framework oriented to the low-resolution gaze-estimation scenario, starting from a level with a lower range of freedom within neural networks (FCNs) to study the weight of certain features (e.g., kappa) when trying to solve the low-resolution gaze-estimation problem. To this end, and with the objective of isolating the contribution of specific features, we will make use of the U2Eyes [23] synthetic database for the extraction of the ground-truth features in an FCN. Although we will not seek to get to the point of understanding how neural networks learn to solve the gaze-estimation problem, we will demonstrate the importance of user calibration in low-resolution systems. In the second part of this work, once the need for calibration has been delimited in a theoretical way, we will study its impact on a system trained with real images. This work is presented as an extension of the work proposed in [24], although deepening the aspects concerning the difference between calibration and non-calibration scenarios. In [24], we studied the effect that pretraining a model using a synthetic dataset has before addressing the gaze estimation calibration in a real dataset, showing that pretraining the model in a similar synthetic domain improves the accuracy when having few images for calibration. However, in [24], we did not include the results without calibration, so we could not study the gain achieved when calibrating.

The main contributions of this work are:A theoretical analysis of the importance of calibration in low-resolution, as well as the necessary features to obtain accurate gaze-estimations in low-resolution.Validation of the role of the calibration in a real environment according to the theoretical framework, extending the work proposed in [24] and analyzing the importance of calibration over other methods that enhance gaze-estimation algorithms.

As the paper presents two differentiated parts, in each of the sections, there will be subsections establishing this distinction. In general, the paper is organized as follows: in Section 2, we will establish the working space, including the databases, networks architecture, data preprocessing and other methodological specifications; in Section 3, we define the experiments conducted for the theoretical framework and the real environment; then, in Section 4, the obtained results will be presented, and then discussed in Section 5. Finally, in Section 6, the conclusions of the work are shown.

## 2. Working Framework

In this section, we will detail both the databases and the methodological specifications adopted during the development of this work. As indicated in the introduction, since this paper deals with two different proposals, each of the subsections will specify the work carried out for each one of the approaches.

### 2.1. Databases

#### 2.1.1. Theoretical Framework

When establishing the theoretical framework to support the need for calibration, we need a controlled environment to isolate the impact of unwanted features. In this work, we have chosen to use an improved version of the U2Eyes synthetic environment [23] that provides a realistic representation of the human eye model. Each synthetic user is characterized by a different face shape (determined by the PCA parameters of the model), a skin texture among 20 different ones and an eye texture, among five options, offered by UnityEyes [25]. In U2Eyes, two new fields were introduced to increase the individuality of each user and incorporate binocular vision dynamics: the horizontal and vertical components of the kappa angles.

The importance of the kappa angles lies in the geometrical description of the human eye. In this description, the optical axis (also called the pupillary axis in the literature) is defined as an imaginary line perpendicular to the cornea that intersects the center of the entrance pupil. In theory, it represents the path followed by a ray of light that enters and leaves the optical system of the eye along the same line. This axis meets the retina missing the fovea, which is the most sensitive zone composed of closely packed cones. For a sharp central vision of any object, the line of sight (that closely approximates to the visual axis or foveal-fixation axis) has to impact the retina just on the fovea so, when gazing at any object, the eyeball rotates to enable this visual axis passing through the entrance pupil center.

Optical and visual axes meet at the nodal point, forming an angle called kappa. A diagram of the kappa angle is depicted in Figure 1. As angle kappa values are relatively small, it is common to neglect the vertical component and even to identify both axes, but such an approach cannot be kept in a gaze-estimation procedure that pursues accuracy features (in degrees) with errors below kappa values. Several studies have been conducted to determine the population’s mean value of the kappa angle. Through different measurement methods, it has been demonstrated that fellow right and left eyes exhibit different kappa values (usually right eyes tend to rotate more than left eyes) and that there is great variability of results arising from different individuals and even among different population groups [26].

In the original version of the U2Eyes environment, a horizontal component value (randomly chosen in the range (3–7) degrees) and a vertical component value (randomly chosen in the range (2–3) degrees) were shared by both eyes, in a symmetrical fashion unable to account for some recent findings: human right and left eyes differ in their axes geometry and vision field; they also differ in dynamic performance in the fixation process; about two-thirds of the population seem to be right-eye dominant; human brain processes information arriving from each eye differently, etc. [28].

For this work, an improved implementation of the kappa parameters belonging to fellow eyes, increasing from two to four fields handled in a non-symmetrical but still in a correlated way for the sake of avoiding infrequent strabismic images, is added. For both eyes, positive values of kappa vertical components mean the visual axis tilts upwards, whereas positive values of kappa horizontal components represent a LoS (Line of Sight) that deviates towards the nasal direction.

A set of four positive values matches the temporal-inferior location of the human fovea regarding the posterior pole of the globe and the fact that both eyes need to rotate in the opposite direction for a clear vision.

It was decided to implement mean and standard kappa values reported in [26] for the Italian population, as is shown in Table 1. This cohort was selected among the three ones included in that work because of its size (343 participants) and similarity between the Italian phenotype and the skin/eyes textures. Full data from these 343 participants, kindly provided by the first author, were statistically processed to derive the covariance matrix and generate a 4-variate normal distribution from where to extract the four kappa angles to each user.

The advantage of using the U2Eyes environment is that it provides labels and landmarks of 2D and 3D features with the certainty that they are correctly and consistently labeled. Relying on previous knowledge about high-resolution, it is known that the LoS depends on the position of the head and orientation of the eyes with respect to it. Based on this knowledge and assuming that, regardless of high or low-resolution, the 3D problem to be solved is the same, i.e., gaze-estimation, the following features provided with the database are proposed as input to the network:Headpose, with position and rotation components.Tags referred to the location of Points of Interest (PoIs). An example is shown in Figure 2a.Landmarks referring to the segmentation of Regions of Interest (RoIs). The RoIs representing the different eye areas are presented in Figure 2b. Both the selected PoIs and RoIs could be extracted from real images, by segmentation tasks (RoIs) or regression (PoIs).Kappa angles, with vertical and horizontal components for both eyes. Together with the PCA coefficients (affecting the head shape) and textures (affecting skin and eyes appearance), these are the labels that capture the individuality of each user, i.e., their unique characteristics. Among these 4 components (PCA, kappa angles and skin/eye textures), we understand that it will be mainly the kappa angles and PCA coefficients that will affect the gaze-estimation for each specific user the most. In other words, these would be the features learned during the calibration stage as they represent an individual’s unique characteristics.

An advantage of using a synthetic environment for image generation is that it allows to generate different subsets controlling the variation of certain features while fixing others. To evaluate the importance of user calibration, specifically by isolating the influence of one of the parameters learned through calibration (kappa angles), the following datasets are created:U2Eyes-Base-20. This is a subdatabase with 20 users, each with its own PCA and skin texture, and sharing eye texture in groups of 4. Each of the users has its own kappa angle values, drawn randomly from the distributions characterized by the means and standard deviations from Table 1. Each user presents 125 head positions, and in each head position looks at 32 + 15 grid points. The gaze angle range is ∼±24°, and the range of the user distances from the grid goes from 35 to 55 cm. This is the range of distances between the eyes and the grid. For more information about the grid points and head pose ranges, see the paper [23].U2Eyes-Kappa-20. It is also a subdatabase with 20 users, twins in terms of PCA conditions, skin texture and eyes with the U2Eyes-Base-20 subdatabase, but in this case all users share the same kappa angle values, these being the mean values in Table 1. Head positions and grid points coincide with the ones in the U2Eyes-Base-20 users too.U2Eyes-Base-300. Similar to the U2Eyes-Base-20 case, this is a database of 300 users, each of them being a unique user (unique combination of PCA, skin and eye textures). Furthermore, the users present their own kappa angles. In this case, the number of head positions is limited to 27 in the same range of distances, although the gaze angle range is the same as in the previous case, using a grid of 15 points.U2Eyes-Kappa-300. As the U2Eyes-Kappa-20 is identical to U2Eyes-Base-20 except for the kappa values of the users, U2Eyes-Kappa-300 is identical to U2Eyes-Base-300 except that all users share the kappa values also used in U2Eyes-Kappa-20.

As we can see, each subdatabase is different in some aspects from the others but allows us to keep certain key aspects common to all of them for better control of the experiments. The reason for using these 4 subsets is the flexibility it provides to train our model using the 300-user variants while keeping the 20-user variants to test the performance of the trained model without the need to establish test and training groups.

#### 2.1.2. Real Environment

The objective of the real environment study is to test the validity of the theoretical framework developed in the synthetic environment in a real scenario.

Ideally, we would have a database of real images in which the kappa values of each user are perfectly known. However, to our knowledge, there is currently no low-resolution database with these characteristics, emphasizing how expensive it would be to create a similar database. The lack of databases for gaze-estimation is a well-known problem, although more and more efforts are being made to create better quality large scale databases [29,30,31,32].

Therefore, this paper aims to extend the experiments proposed in [24], where the I2Head database [32] was used because of the similarity of the images in relation to the images of the U2Eyes environment. The I2Head dataset is a public database of 12 users. This dataset uses two grids, one of 17 points and the other of 65 points that the user has to lookat while being recorded. In total, for each user, 6 sessions for the 17-grid and 2 sessions for the 65-grid were recorded, with 10 images per point, which gives an amount of 28,840 images for the complete dataset. In [24], head centered sessions were used, i.e., 2 sessions of 17 and the 2 sessions of 65 points (164 images). For more information about the dataset, please refer to [32].

The calibration stage we proposed [24] resembles the paradigm used for high-resolution systems in which a pre-established model is adapted to the subject while a grid of points is shown in the screen. In the previous work, we performed this process, but did not indicate which results were prior to calibration. By adding these pre-calibration results, it is possible to measure the impact of the calibration by comparing the obtained results with the theoretical framework.

### 2.2. Data Processing

#### 2.2.1. Theoretical Framework

For the training of the models, it was decided to use the subdatabases of 300 users for training and the subdatabases of 20 users for testing. After selecting the features of interest to be used, these were scaled to fall in the range −1 to 1, so that all the features would be in the same range, centered around zero and preserving their original distribution. It should be noted that this is a scaling and not a normalization procedure, so this process does not ensure that the mean has a value of zero. The scaling is therefore calculated as follows:(1)$$X_{Scaled}=\frac{X-X_{Min}}{X_{Max}-X_{Min}}*\left(MaxRange-MinRange\right)+MinRange$$
where *X* is the feature to be scaled, $X_{Min}$ and $X_{Max}$ are the minimum and maximum values of that feature in the training dataset, and MinRange and MaxRange delimit the final range, in this case, −1 to 1.

For the calculation of this scaling, only the values of the characteristics present in each of the training subdatabases are considered. In this way, we ensure that the subdatabases used for testing the models are effectively isolated.

An aspect to be taken into account falls on the components related to the kappa angles and the calculation of the scaler based on the U2Eyes-Kappa-300 subbase. According to the definition given in Section 2.1.1, the Kappa subbases are characterized because the kappa angles values are the same for all the users in the dataset. If we were to apply Equation (Equation 1) in this case, we would be facing an indeterminate form, 0/0. To avoid this case, different solutions were examined:Altering the input to the network to use 4 fewer parameters;Applying a uniform random value, in the range −1 to 1, to the values of these features for all users;

Since we seek to compare in the most transparent way the training cases between the use of the U2Eyes-Base-300 and U2Eyes-Kappa-300 subbase, it would be desirable to keep exactly the same network shape for the two cases. Therefore, it was decided to assign uniform random values in the range −1 to 1 to the users in Kappa-subbases. Assigning random values externally to the rest of the parameters would break any kind of dependence that may be existing between these parameters and the look-at-point that the network has to estimate. Nevertheless, as the training progresses, the network learns to ignore these features.

#### 2.2.2. Real Environment

Following the steps described in [24], the same preprocessing was performed on the I2Head images. The process is detailed in Figure 3 and consists of rotating and cropping the images in such a way that the roll component is eliminated, and all images are guaranteed to have the same size by adding padding.

### 2.3. Networks Architecture

#### 2.3.1. Theoretical Framework

The network architecture is based on FCNs and consists of the following topology:An input layer of 366 inputs, equivalent to the number of features employed;Six layers with 512 neurons per layer, ELU activations, using the He uniform variance scaling initializer [33], and kernel regularization l2 with value 0.01 per layer;A final layer of 2 neurons, with linear activation, which returns the x and y components of the estimated gaze;

A visual description of the network is shown in Figure 4.

#### 2.3.2. Real Environment

The network design is the same as in the original paper [24]. The input to the network consists of three-channel (RGB) images with a size of 388 × 84. A Resnet-18 is used as the backbone, followed by a Global Average Pooling layer and three Fully Connected layers with 645, 32 and 16 neurons and ReLU activation. Finally, as for the theoretical framework, the network output consists of a last fully connected layer with linear activation and 2 neurons to return the gaze-estimation (x and y). Similarly to the previous work [24], the training dynamics between the weights of the Resnet-18 trained on ImageNet used for the backbone [34] and the network trained on the U2Eyes database were kept. The results obtained over ImageNet ensure that the network is able to extract meaningful features from images. In this way, we can also analyze whether pretraining on a similar domain has an impact on the results obtained by the calibration. The topology of the network is shown in Figure 5.

### 2.4. Implementation Details

This subsection will explain the details that allow the reproducibility of the work presented. In any case, the code used for the experiments will be published openly on the Github platform.

#### 2.4.1. Common Characteristics

Regardless of the architecture, database used, preprocessing and other characteristics that differentiate the part of the theoretical framework from the real implementation, the essence of both parts is that they try to predict the direction of gaze or look-at-point, understanding it as a regression task. In both cases, the neural networks will return two continuous values, x and y coordinates, from the inputs. Therefore, for the training of both systems we find some common elements. One of these elements is the minimization of the loss function. In both cases, the loss function is the Euclidean distance between the look-at-point estimated by the network and the real look-at-point. The loss function follows the next equation:(2)$$Loss:=\frac{1}{N}\sum_{i=1}^{N}{∥p-\hat{p}∥}_{2},$$
where *p* is the real look-at-point, (*x*, *y*), $\hat{p}$ is the estimated look-at-point ($\hat{x}$, $\hat{y}$) and *N* is the number of images per batch.

Another common element in both setups is the presentation of the results. By the very nature of the problem, analyzing only the difference in distances between the estimated look-at-point and the real look-at-point would be useless if the distance at which the subject is located is not taken into account. That is, the same error distance is not equivalent for two subjects placed at different distances. Therefore, it is common to analyze the results in terms of the angular error made during the estimation. Since in our case both databases collect the distance of the user to the grid of points they have to look-at, the transformation between the estimation error and the angular error can be performed using the following formula:(3)$$AngularError:=\arctan(\frac{∥p-\hat{p}{∥}_{2}}{d})$$
where *p* is the real look-at-point, (*x*, *y*), $\hat{p}$ is the estimated look-at-point ($\hat{x}$, $\hat{y}$) and *d* is the distance from the grid. It is assumed that the distance between user and grid corresponds to the major leg of a right triangle, while the minor leg corresponds to the estimation error. Thus, the error angle corresponds to the angle formed by the hypotenuse and the major leg. A more graphic explanation can be found in Figure 6.

#### 2.4.2. Theoretical Framework

For the different experiments in the theoretical framework, an identical training configuration was used, although the ideal case would be to adapt some of the training conditions, such as the value of the initial learning rate or its decay, to the different training cases, depending on whether we use the U2Eyes-Base-300 or U2Eyes-Kappa-300 database. However, although we accept that the values that we will present below are not necessarily the optimal values for training in both databases, they allow us to state that the only conditions that change between the two training cases are the databases. This is a compromise solution, as in the case of keeping the number of features constant by randomizing the values of U2Eyes-Kappa-300 to preserve two similar configurations as much as possible.

For training, the Adam optimizer is used, with a learning rate of $1\times 10^{-4}$ for the first 1600 epochs, $1\times 10^{-5}$ for the next 1200 epochs and $1\times 10^{-6}$ for the final 200 epochs.

The more relevant hyper-parameters while training this network are the learning rate and the number of epochs. The Adam optimizer is a popular choice for training neural networks. The initial learning rate value is chosen following the method proposed by [35]. For this method, the learning rate is progressively increased while checking the obtained loss value. When the network stops learning, which is determined by the moment the loss value remains the same for some iterations and then slowly starts increasing, we stop the process and select the initial learning rate as a tenth of the learning rate value at that moment. The number of epochs was chosen empirically after some test trainings. Each number of epochs corresponds with the moment the network stopped learning for a given learning rate. At that point, the learning rate was reduced to a tenth of the value, and then the training was resumed until the next plateau in the training. We repeated this method twice. To avoid overfitting as a result of this process, we utilized L2 regularizers in the layers of the network.

#### 2.4.3. Real Environment

Following the characteristics of the original paper [24], the Adam optimizer was used to train the experiments, with a cyclic triangular learning rate schedule based on the work of [35], with a maximum value of $2\times 10^{-3}$ and a minimum value of $2\times 10^{-4}$. The networks are trained for 240 epochs, of which the first 200 use the learning rate policy detailed above, and in the final 40 epochs, the learning rate is progressively reduced until the minimum value of $2\times 10^{-5}$ is reached in epoch 240. In this case, we do not set a specific batch size, since due to the nature of the original experiments, the volume of input data was changing as more or less users were added to the training, but we ensure that each epoch consists of 128 training steps.

As an extension of the original paper, the hyper-parameters have remained the same. In the original paper, the learning rate was chosen with the same method described in Section 2.4.2, although in this case the learning rate schedule was a cyclic learning rate, as described in [35]. The learning rate schedules comprehend a series of techniques that prevent the networks from getting stuck during training.

## 3. Experiments

As in the previous sections, for the definition of the experiments, we will focus on each of the two blocks separately to facilitate the interpretation of the study.

### 3.1. Theoretical Framework

The experiments carried out to test the theoretical framework consist of training two neural networks, which were detailed in Section 2.3.1. The first of the trained networks uses the U2Eyes-Base-300 database, while the other network employs the U2Eyes-Kappa-300 database. Since the images generated in the former, and hence the look-at-points, are calculated using different values of kappa angles, a properly trained network will have to learn to interpret these values to return a correct estimation. In the case of using the U2Eyes-Kappa-300 database, since all images have the same kappa angle values, the network learns to prescind from them when estimating the gaze.

After training the networks, each of them is tested on the two complementary databases, U2Eyes-Base-20 and U2Eyes-Kappa-20. Both networks are tested on both databases. In the case that the network would be trained correctly, the expected results of these experiments would be:The network trained on the U2Eyes-Base-300 database should return similar results when tested on either database, since it should learn to interpret the kappa angle values to solve the gaze-estimation.The network trained on the U2Eyes-Kappa-300 database should return correct results when tested on the U2Eyes-Kappa-20 database, since both use the same kappa values and the network has learned to solve the gaze-estimation problem for these values, and significantly worse results for the U2Eyes-Base-20 database, since it does not understand the influence that other kappa values have on gaze-estimation.

### 3.2. Real Environment

As for the real part, starting from the work presented in [24], we extend the presented experiments by including the cases where no calibration was used.

In [24], we studied the impact that the number of training images have in a gaze-estimation model depending on if the model was pretrained over synthetic eye images or using the weights from ImageNet [34], a general purpose computer vision database. While training, the model was always fed with at least 34 images from a test user, simulating a calibration process. However, during the original experiment it was not checked what results were obtained when no calibration images were fed, i.e., if no calibration was performed. Therefore, it was not possible to isolate the impact of the calibration over the importance of pretraining the model with a similar or different domain. With this extension, we will experiment without calibration so we can analyze its importance by comparing with the original results.

In [24], experiments were presented to observe the impact of pretraining a network using a synthetic dataset whose domain is closer to gaze-estimation before facing a real dataset. For this, a model that uses a Resnet-18 as a backbone directly with weights from ImageNet was compared against the same model but previously pretrained over the U2Eyes dataset. Furthermore, the experiments were configured so that the importance of the number of images while calibrating a model for gaze estimation could be measured. To do this, we used 34 images of the user to be calibrated (the two centered sessions of 17 points of the I2Head dataset) that were included in the training set together with a varying number of additional users from the I2Head dataset, ranging from 0 users (only calibration) to 11 users. Each one of the users in training added 130 additional images to the training dataset. The model was then tested using the remaining two centered sessions of 65 points of the calibrated subject. A summary of the experiments configuration is shown in Table 2.

To extend our work ([24]), we propose to emulate the past experiments with a similar configuration that is detailed in Table 3 but without using any calibration. This way, we could measure the difference in the gaze-estimation performance with and without calibration. The fact of training by differentiating between using the pretrained network in U2Eyes or the ImageNet weights is maintained too, in order to obtain complementary information about the impact of using a pretrained network in the target domain and to evaluate the effects of calibration in both scenarios.

### 3.3. Limitations of the Experiments

Due to the nature of the proposed experiments, they are subject to some limitations:The theoretical framework experiments, as it uses a synthetic framework, have perfectly annotated and controlled features. However, in a real environment, these characteristics present noise due to the labeling process, wherever it is automatic or manual. This added noise will have an effect on the final gaze estimation. To overcome this limitation, it would be necessary to perform an analysis to characterize the noise, so it could be added to the synthetic features.In the real environment, we work with a specific dataset where the head motion and boundary conditions (illumination, distance to the grid, static background, etc.) are controlled, as opposed to a “in-the-wild” scenario.

## 4. Results

After defining the training process and the configuration, we present the obtained results. As in the other sections, this section is divided into the theoretical framework and the real environment.

### 4.1. Theoretical Framework

For the analysis of the results of the experiments regarding the theoretical framework, we present in Figure 7 the boxplot distributions of the angular error for the different users and experiments configuration. Each of the panels of the figure represents one of the two training configurations (training on the U2Eyes-Base-300 and U2Eyes-Kappa-300 database) tested on one of the test databases (U2Eyes-Base-20 and U2Eyes-Kappa-20). The x-axis shows the user’s number to which the boxplot corresponds out of the 20 possible ones. On the vertical axis, as explained in Section 2.4.1, the gaze-estimation error of the model measured in degrees is represented. To facilitate the comparison between the figures, the range of the vertical axis is the same for all representations.

In addition, Table 4 shows a breakdown of some statistical components of interest (mean, standard deviation, median, 25% and 75% quantiles and maximum and minimum values) of the error obtained on the 11,500 samples that make up the test databases. Figure 8 represents the means errors for each test in a visual comparison.

### 4.2. Real Environment

As in the previous case, we have used both a numerical representation in the form of a table (Table 5) and two graphical solutions in the form of boxplots (Figure 9 and Figure 10) to present the results of the experiments.

As this is an extension of the work done in [24], to facilitate the interpretation of the results, Figure 9a shows results for the two training models while increasing the number of users in model training. It corresponds to the original experiment, with calibration. The X axis shows the number of users used while training that were not the calibration user, so in Figure 9a, the real number of users present while training for each case is the value of the X axis plus one, i.e., the calibration user. Figure 9b presents the same distribution but for the experiment without calibration. In this case, as there is no calibration process, the real number of users used while training corresponds with the X axis.

Figure 9c,d shows a direct comparison between the results obtained with and without calibration for the U2Eyes training and for the ImageNet training. As in the previous figures, the Y-axis represents the estimation error measured in degrees, and the X-axis shows the number of users present in the training, without considering the calibration user. The estimation error is the average error obtained when training with different users for testing.

Likewise, Table 5 shows the mean and median values obtained for each of the training configurations. As in Figure 9, the number of users in training goes from 0 (only calibration user, for the experiments of the original paper) to 11 (all additional users are used in training). In the table, for ease of reading, the maximum and minimum values obtained in each column are marked in bold.

In Figure 10, the results from the four experiments, the two experiments from the original paper and the two extensions without calibration from this work are shown without differentiating by the number of users/images used while training.

## 5. Discussion

### 5.1. Theoretical Framework

For the analysis of the results, we will first check whether the hypotheses we put forward in Section 3.1 were correct:


*The network trained on the U2Eyes-Base-300 database should return similar results when tested on either database, since it should learn to interpret the kappa angle values to solve the gaze-estimation*


Observing the values presented in Table 4 and Figure 7a,b, we can affirm that when effectively training on the U2Eyes-Base-300 database, the results obtained are similar regardless of the database on which they are tested (U2Eyes-Base-20 and U2Eyes-Kappa-20).


*The network trained on the U2Eyes-Kappa-300 database should return correct results when tested on the U2Eyes-Kappa-20 database, since both use the same kappa values and the network has learned to solve the gaze-estimation problem for these values, and significantly worse results for the U2Eyes-Base-20 database, since it does not understand the influence that other kappa values have on gaze-estimation*


We can also affirm that this premise is fulfilled. The results obtained when testing on U2Eyes-Kappa-20 (Figure 7c) are the best among the four tests, although with values close to those obtained when training the model with the U2Eyes-Base-300 database (Figure 7a,b). However, the values obtained when testing on U2Eyes-Base-20 (Figure 7d) present a high disparity when compared to the rest of the results. This fits with the expected behavior, since the trained network is not able to take into account the influence of the kappa angles in the gaze value.

If we compare results obtained when calibration data is added (0.7° in mean when training over U2Eyes-Base-300 and testing over U2Eyes-Base-20) versus results obtained when the network does not use any calibration (2.7° in mean when training over U2Eyes-Kappa-300 and testing over U2Eyes-Base-20), the improvement is 75%, even if the calibration would be limited to learn only the kappa angles. To quickly understand the implications of this enhancement, we can refer to Figure 8. If we adjusted the size of the black circle (angular error of 2.7°) to a diameter of 2.2 centimeters, the circles would correspond with the mean error for a gaze-estimation at a distance of 45 centimeters, which could be the distance from a user to a computer screen while working. Depending on our target application, a correct calibration may involve the usability or not of the gaze-estimation solution.

Another interesting aspect to highlight in Figure 7d is that the inter-user angular error is highly variable, but the variance of the error for each user is small, i.e., for each user there is a tendency to make a certain estimation error. This opens the door to try to find a relationship between the kappa values of these users and the variance of the obtained error.

In addition to the confirmation of the initial hypothesis, the results reveal an interesting outcome. In the closest case of a “real” situation, which would correspond to training using the U2Eyes-Base-300 database and testing on U2Eyes-Base-20, we observe a value of 0.7° of error on average, close to the ∼0.5° of error present in high-resolution commercial solutions. Moreover, given that the training conditions, such as the effective choice of hyperparameters, were not the most critical aspect of this work, we are optimistic that this error could still be further reduced.

The presence of other unique characteristics of each individual may affect the results, although to a lesser extent compared with the kappa angles, which would explain why the results obtained in Figure 7a–c are not equal for all users, and that for certain specific users (e.g., users 4, 9 and 18) the errors are bigger than for the remaining ones.

These results emphasize that calibration processes are important and enhance the results obtained in gaze-estimation. This was already known from high-resolution scenarios. The main novelty from these results is that we have established a theoretical framework for low-resolution calibration that, although it learns from high-resolution previous knowledge, it is a different entity by itself. This theoretical framework has even set a reference accuracy, 0.7º of error on average, that only depends on images from a low-resolution scenario, probing that we can further improve the results from low-resolution gaze estimation. Furthermore, in view of the nature of the features used, it would be necessary a paradigm shift in the design of the models, especially for the models based on end-to-end CNNs.

When we talk about the paradigm shift, we propose that it seems reasonable to use specialized architectures and error functions to achieve end-to-end CNN models able to learn how to extract relevant information from the eye/face images for the gaze-estimation computer vision task under study. The importance of full-face images is discussed in other works [36] that concluded that by providing the network with additional information other than the eye region improves the performance of the gaze-estimation. In [29], it was concluded that by carefully selecting the right features, not only the eye region, their method could generalize better to other databases in addition to the one used for training. The need for more complex architectures to perform different tasks simultaneously can be grasped from [37], where two neural networks were used, one to model the head pose and the other to analyze the eyeball movement. The results from our experiments corroborate and extends this key idea: gaze-estimation is a complex process where multiple parts interact with each other: region of interest segmentation, pose estimation, points of interest regression, etc., and each of them is the result from different computer vision tasks, so an effective gaze-estimation model has to tackle and solve all of them simultaneously. This work has tried to identify some features that could be extracted to obtain accurate results. Our work opens the possibility of studying which minimum theoretical features in low-resolution could obtain results close to the ones obtained by high-resolution systems, which would allow to reduce the number of necessary features and, therefore, to simplify the design of gaze estimation models.

### 5.2. Real Environment

In view of the results obtained from the experiments, the first thing to note at a general level from the values in Table 5 is that, regardless of whether we use a training strategy based on approaching the domain (U2Eyes) or using a pretrained general purpose backbone (ImageNet), the influence of calibration on the final results is significant, with improvements of around 50% in the most favorable cases (training with as many additional users as possible). The need for calibration to reduce the error below 2.5° for either of the two training configurations is clear from the table.

Moving on to more particular cases, in the original paper [24] we concluded that, to compensate for the lack of useful images during the training process, the models benefit from pretraining over a dataset whose domain was close to the final training scenario (eye images). However, from our results, we have noticed that this behavior is affected by whether calibration is used or not. Although the U2Eyes pretrained model still outperforms the ImageNet model when training on few images (Figure 9b n° of I2Head users in training from 1 to 3 users), the difference is not as large as when calibration was performed (Figure 9a n° of users in training from 1 to 3 users). This can be better observed in Figure 10. While the boxplots from the experiments with no calibration are similar, the boxplots of the original experiments, calibrated to the test user, show a difference when pretraining over U2Eyes or when using the weights from ImageNet, knowing from Figure 9 that the main impact of this pretraining step affects the scenarios with lack of training images.

The trend observed in the original experiment where, as the number of users increased, the model with the Resnet-18 training weights on ImageNet tended to obtain better results than the model pretrained on U2Eyes, is confirmed. The main difference is that the obtained results using calibration exceed the ones in the no calibration scenario. It can be observed in Table 5 that the ImageNet model slightly outperforms the U2Eyes model, as the obtained results for both the mean and the median are better in the case of ImageNet when the training users go from 4 to 11 in a no calibration scenario compared with the U2Eyes strategy.

## 6. Conclusions

The experiments carried out in this paper follow two finalities, as it has been maintained throughout the document:To try to establish a theoretical framework that shows the influence and impact of calibration in the cases of gaze-estimation for low-resolution, understanding the calibration for fitting the gaze-estimation model to individual and intrinsic conditions of the user. In our case, determined by the kappa parameters of the synthetic environment U2Eyes;To check the impact of this calibration on a real database.

To achieve these objectives, two experimental setups have been employed. For the theoretical framework, a fully connected neural network was trained using selected features from a synthetic environment based on our previous knowledge about high-resolution systems. From the synthetic environment, four datasets with controlled characteristics were defined and used. For the real environment, we extended the work proposed in [24] to compare the results obtained with and without calibration.

From these experiments, we extract three main conclusions.

The first conclusion of this work is that user calibration is shown to play a key role in reducing gaze-estimation error. Improvements of 75% in the theoretical framework and 50% in the real environment demonstrate that, as was the case in the high-resolution environments, calibration is also confirmed as a key process in low-resolution gaze-estimation.

The second conclusion shown is that, theoretically, values close to the ones obtained by high-resolution (0.7°) can be obtained for gaze-estimation with a careful choice of features and by employing calibration processes. Although this is a conclusion drawn on a synthetic database, we believe that there is room for improvement in the accuracies obtained by low-resolution systems.

This second conclusion leads to a third conclusion, which states that when building end-to-end models for gaze-estimation, we must think of them as the sum of several computer vision tasks, which can increase their complexity, and processes in which it will be necessary to add user calibration stages.

Finally, after the experiments extending the work in [24], we can conclude that the choice about the use of a calibration process can outshine other promising approaches. For example, without the use of calibration, the benefits of pretraining a model in a closer domain to compensate the lack of useful gaze data images instead of in a more general computer vision dataset would not be as relevant. Although in this paper there has been only one enhancement technique analyzed (pretraining in a similar domain), it is possible that this conclusion could be extended to other procedures, i.e., the lack of a calibration process could overshadow other methods that could be used to improve the gaze-estimation results.

## Figures and Tables

**Figure 1 sensors-21-05109-f001:**
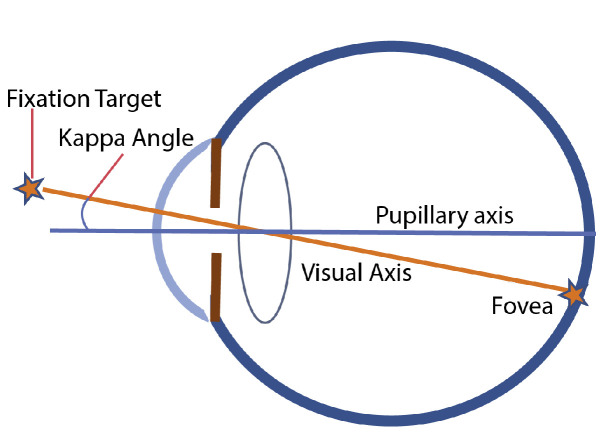
Axial diagram with an approximation of the human eyeball model. The image corresponds with a top-view of a left eye. Considering a fixation target, we could distinguish between the visual axis and the pupillary axis. The kappa angle establishes the relation between the two axes. Original image from [27].

**Figure 2 sensors-21-05109-f002:**
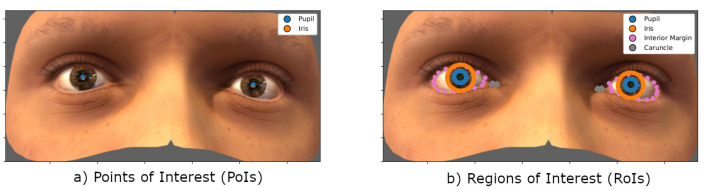
Example showing the correspondence of the features used to establish the methodological framework over an image from the U2Eyes. (**a**) Projection of 3D center of pupil and iris. Points of Interest (PoIs) of the image. (**b**) Segmentation of the pupil, iris, interior margin and caruncle for both eyes. Regions of Interest (RoIs) of the image. Images from U2Eyes database [23].

**Figure 3 sensors-21-05109-f003:**
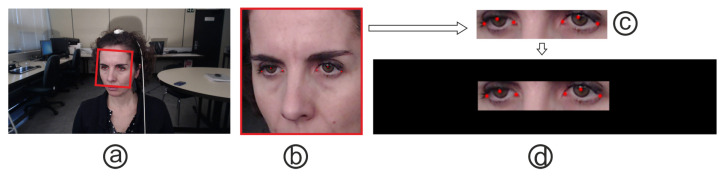
Image preprocessing used in the real environment experiments. The images (**a**) are rotated (**b**) so that the eyes are in the same line and cropped to the bounding box (**c**), which contains as meaningful information as possible. Then, a padding process (**d**) assures that all images have the same size before feeding them to the network. Images from the I2Head database [32].

**Figure 4 sensors-21-05109-f004:**
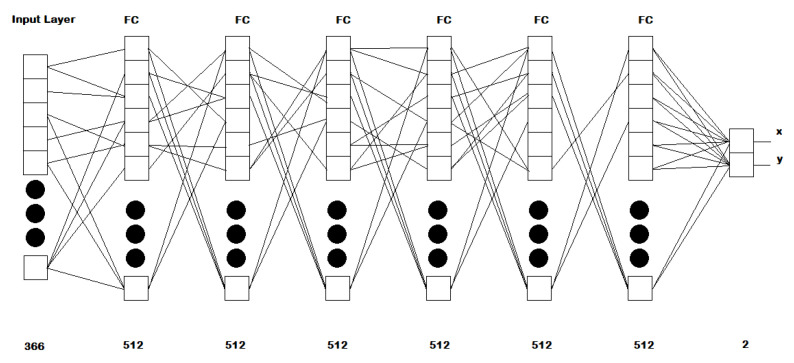
Architecture proposed for the methodological framework. The network consists of an input for 366 features, followed by 6 fully connected layers with 512 neurons each, and a final fully connected layer that outputs the gaze components (x and y).

**Figure 5 sensors-21-05109-f005:**
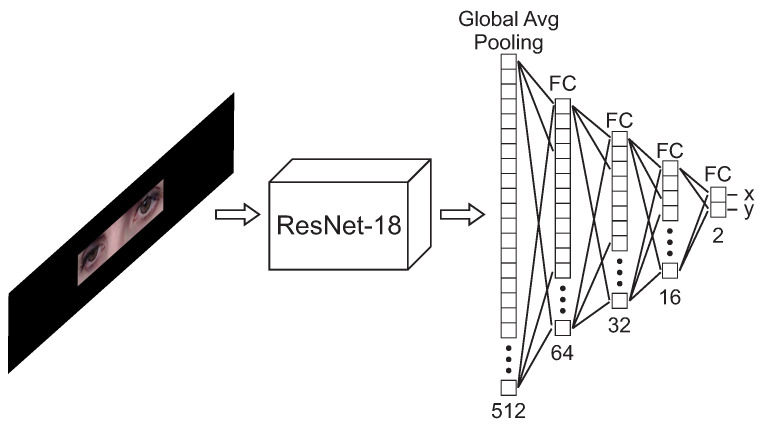
Architecture proposed. The backbone consists of a Resnet-18 to extract meaningful characteristics from the image. Then, these characteristics are fed into a fully connected regressor network to obtain the final gaze components. Image used as input to the network from the I2Head database [32].

**Figure 6 sensors-21-05109-f006:**
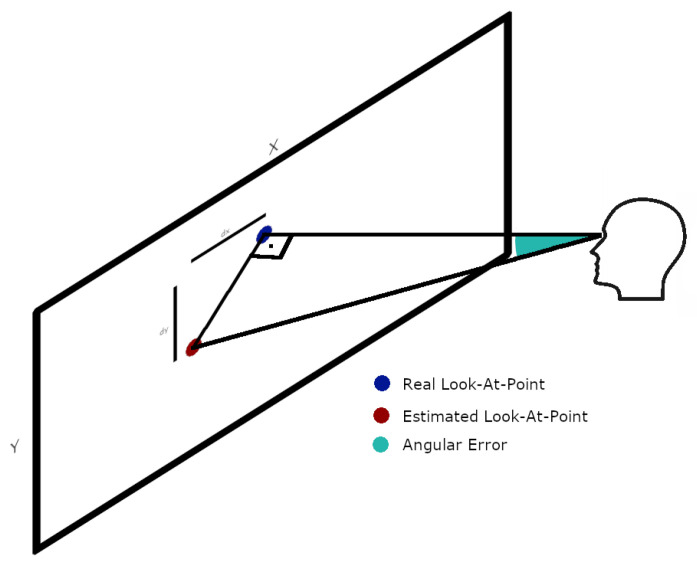
Angular error resulting from gaze-estimation. The angular error corresponds with the angle between the vectors Eyes Position—Real look-at-point and Eyes Position—Estimated look-at-point.

**Figure 7 sensors-21-05109-f007:**
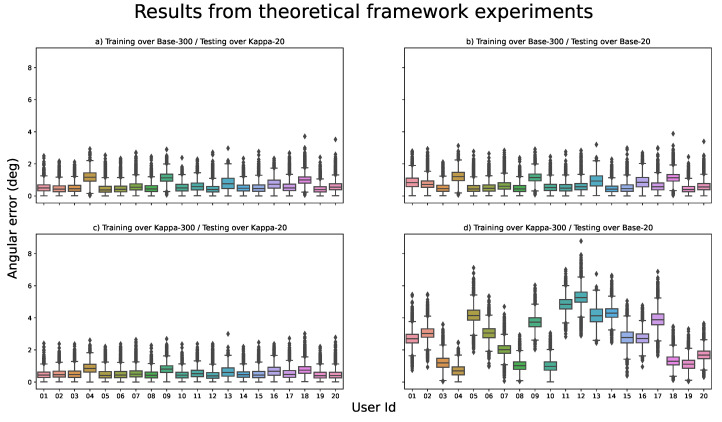
Boxplot representations of the results obtained for the theoretical framework experiments. The figures share the same X and Y axes. The X axis corresponds to the user Id from which each boxplot is computed. The Y axis is the angular error in degrees. (**a**) Results obtained when training over U2Eyes-Base-300 and testing over U2Eyes-Kappa-20. (**b**) Results obtained when training over U2Eyes-Base-300 and testing over U2Eyes-Base-20. (**c**) Results obtained when training over U2Eyes-Kappa-300 and testing over U2Eyes-Kappa-20. (**d**) Results obtained when training over U2Eyes-Kappa-300 and testing over U2Eyes-Base-20.

**Figure 8 sensors-21-05109-f008:**
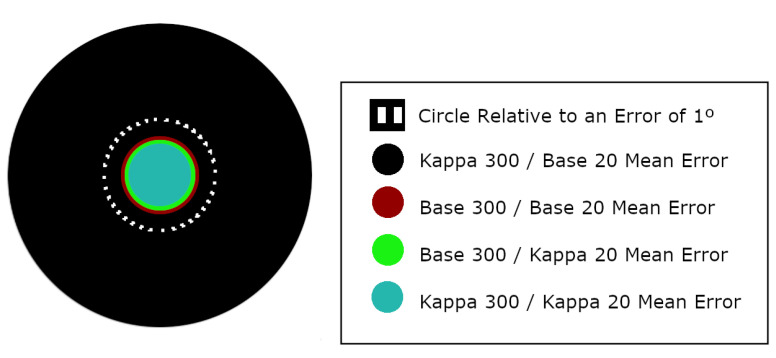
Visual comparison of the mean results presented in Table 4. The black circle is the equivalent representation of an error of 2.749° (model trained over Kappa-300 and tested over Base-20); the brown circle, an error of 0.709° (Base-300/Base-20); the green circle, an error of 0.639° (Base-300/Kappa-20); and the blue circle, an error of 0.566° (Kappa-300/Kappa-20). As a base reference, the dotted white lane represents an error of 1°.

**Figure 9 sensors-21-05109-f009:**
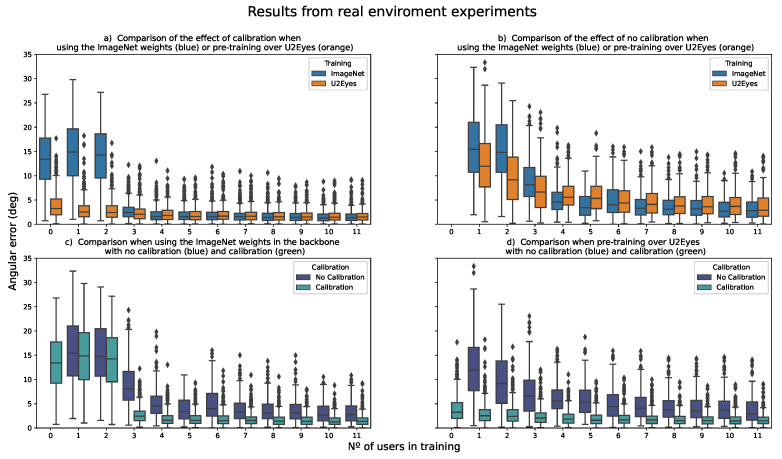
Results of the experiments from the real environment. (**a**) Original experiments from the paper [24]. Training with calibration. We can observe how the model pretrained over U2Eyes outperforms the ImageNet model when the number of users in training is small. (**b**) Results obtained when training the networks (U2Eyes and Imagenet) without calibration. Same configuration as in the original paper but without calibration. In this case, the advantage of pretraining over U2Eyes when the number of users in training is small is not as significant as in the previous case. (**c**) Comparison of the results obtained from the original experiment (calibration) and the new experiment (no calibration) when training over the model with the ImageNet weights. (**d**) Comparison of the results obtained from the original experiment (calibration) and the new experiment (no calibration) when training using the model pretrained over U2Eyes.

**Figure 10 sensors-21-05109-f010:**
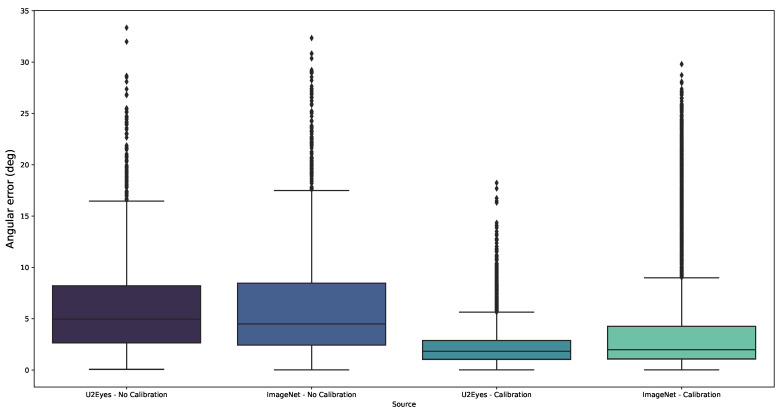
Another view of the results from the real environment. The angular error results for each training configuration are shown in a boxplot. The first two boxplots correspond with the experiments carried in this paper, while the third and fourth boxplot corresponds with the original experiments [24]. In the figure, when no calibration process is performed, there is almost no difference between pretraining over ImageNet or U2Eyes; however, when using calibration, the difference between the two training strategies is relevant.

**Table 1 sensors-21-05109-t001:** Visual axis vertical and horizontal tilts (in degrees).

Kappa Angle	Horizontal Right	Vertical Right	Horizontal Left	Vertical Left
Mean	5.4237	2.512	2.1375	4.4458
Std	2.4978	2.6782	1.9672	2.7185

**Table 2 sensors-21-05109-t002:** Original configurations of the different experiments according to the training mode, U2Eyes or ImageNet, and the number of users and images in the training and testing phase. The K parameter varies from 0 to 11, i.e., 0 indicates that none of the subjects of the I2Head dataset has been included to train the model, except for the subject to be calibrated, and 11 indicates that all the additional subjects are present in the training phase. For the calibration, 34 images of the subject are employed while the test is conducted over 130 subject images unseen by the model.

Model	Train # Users/Images	Calibration # Users/Images	Total Train # Images	Test # Users/Images
ImageNet	K/K × 130	1/34	K × 130 + 34	1/130
U2Eyes	K/K × 130	1/34	K × 130 + 34	1/130

**Table 3 sensors-21-05109-t003:** New configurations of the different experiments according to the training mode, U2Eyes or ImageNet, and the number of users and images in the training and testing phase. In this new case, the K parameter varies from 1 to 11, as zero was reserved when training only with the calibration user. Additionally, there is no “Calibration #Users/Images" column as calibration is not conducted. The only images in the training dataset comes from users different from the test user.

Model	Train # Users/Images	Total Train # Images	Test # Users/Images
ImageNet	K/K × 130	K × 130	1/130
U2Eyes	K/K × 130	K × 130	1/130

**Table 4 sensors-21-05109-t004:** Angular error (in degrees) of the gaze-estimation for the theoretical framework. The first row determines the dataset for which the model was trained (U2Eyes-Base-300 and U2Eyes-Kappa-300) and the second row the dataset used to test the trained model (U2Eyes-Base-20 and U2Eyes-Kappa-20).

Training Database	U2Eyes-Base-300		U2Eyes-Kappa-300	
Testing Database	U2Eyes-Kappa-20	U2Eyes-Base-20	U2Eyes-Kappa-20	U2Eyes-Base-20
N° Samples	117,500	117,500	117,500	117,500
Mean	0.639	0.709	0.566	2.749
Std	0.393	0.419	0.342	1.462
Min	0.001	0.001	0.001	0.008
25%	0.342	0.390	0.311	1.399
50%	0.556	0.628	0.501	2.753
75%	0.866	0.963	0.753	3.927
Max	3.716	3.885	3.014	8.788

**Table 5 sensors-21-05109-t005:** Results from the different experiment configurations. Mean and median angular errors in degrees are shown for each case. The number of users in training represent the number of subjects that participate during the training that are not test users. The maximum and minimum values for each one of the columns are emphasized.

	Mean				Median			
Model	ImageNet	ImageNet	U2Eyes	U2Eyes	ImageNet	ImageNet	U2Eyes	U2Eyes
Calibration	Yes	No	Yes	No	Yes	No	Yes	No
Users in training								
0	13,615	None	**3891**	None	13,401	None	**3243**	None
1	**14,812**	**16,133**	2967	**12,656**	**14,880**	**15,446**	2522	**11,935**
2	14,069	15,456	2861	9982	14,255	14,807	2404	9169
3	2710	9160	2376	7387	2426	8126	2061	6626
4	1918	5278	2106	6158	1598	4561	1814	5572
5	1934	3998	191	5893	1586	3389	1599	5326
6	1883	4940	1963	4945	1541	4001	1672	4394
7	1770	3853	1887	4726	1544	3310	1648	4090
8	1660	3773	1784	4367	1397	3098	1535	3736
9	1621	3797	1748	4120	1384	3182	1510	3582
10	**1541**	**3261**	1723	4260	**1267**	**2656**	**1446**	3681
11	1559	3302	**1714**	**3791**	1344	2770	1486	**2866**

## Data Availability

For more information about the databases, please refer to: http://www.unavarra.es/gi4e (accessed on 27 July 2021) or to the correspondence author.

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
