# Peer review of "Low-Cost Eye Tracking Calibration: A Knowledge-Based Studyâ€"

_sensors, 2021, doi:10.3390/s21155109_

Round 1
Reviewer 1 Report
This work is investigating the effect of calibration and other factors for ml based eye tracking in contexts where high precision devices may not be available. It shows e.g. that even for low resolution eye tracking, calibration can significantly increase tracking accuracy. This is not unexpected, but it's analysis and quantification are valuable.
The paper is well written and structured . The paper provides detailed insights on the quality of ml based eye tracking, a highly relevant area in the current times as the authors well motivate. Overall I'm positive on this work. The authors can consider the following points to further strengthen their manuscript.- I'd recommend further proofreading (less on typos, more on reading flow) - A significant problem of calibration in low resolution scenarios like webcam/Laptop or mobile device is usability. It is often difficult to collect accurate calibration data. To this end, more flexible approaches have been proposed and studied, which may be of interest to discuss on this paper. For example, the following approaches could make it easier to collect such data, without needing assistance by a supervisor. -- PursuitCalibration, Pfeuffer et al., UIST '13 -- Time and space efficient calibration, Drewes et al., ETRA'19 - The methodology of using ML should be somehow easier described. It is currently still quite difficult to comprehend the overall use of methods. What (hyper)parameters were relevant? How were they tested? Were defaults used? How does the use differ from prior work? - Besides, I think the content of the paper with the analysis and report on ml based eye tracking is sound, although I am not an expert in this area. The results are making sense, and the many variables analysed provide further insights into the feasibility of the tested methods. To further strengthen the work, I believe discussing the findings in relation to prior work would be valuable. For example, the recent work on using mobile eye tracking with ML by Sugano and Bulling (or others) could be an interesting baseline. - _All_ the main findings/conclusions, such as that calibration improves low-resolution eye-tracking, should be discussed relative to their novelty. How novel is it to find that out? How much would it have been expected that this result is found; how well do we better understand the issues involved with it? - As the authors mention their work build up on [19], a clearer distinction of the work to the prior would be useful, too. -
Author Response
Dear reviewer,
I hope this email finds you well,
Please find attached the response to your review.
Thank you for your time and insightful comments on our
paper,

Reviewer 2 Report
1) The inspiration of your work must be highlighted. 2) More experimental figures results are welcome 3) Conclusions section need more explanations and discussion the resultsAuthor Response
Dear reviewer,
I hope this email finds you well,
Please find attached the response to your review.
Thank you for your time and insightful comments on our
paper,

Reviewer 3 Report
Major Comments
- What is the motivation of the proposed work?
- Introduction needs to explain the main contributions of the work clearer.
- The novelty of this paper is not clear. The difference between present work and previous Works should be highlighted.
- Authors must explain in detail the introduction section.
- Authors must develop the framework/architecture of the proposed methods
- There is need of flowchart and pseudocode of the proposed techniques
- Proposed methods should be compared with the state-of-the-art existing techniques
- Research gaps, objectives of the proposed work should be clearly justified.
- To strengthen the Intro and related work sections authors are highly recommended to add the high quality manuscripts < ‘On the Security and Privacy Challenges of Virtual Assistants’, Sensors,21, No. 7, pp.2312, 2021>, < Machine Learning Meets Communication Networks: Current Trends and Future Challenges’, IEEE Access, vol.8, no. 2020, pp. 223418 - 223460, 2020 >, <‘A review on 802.11 MAC protocols industrial standards, architecture elements for providing QoS guarantee, supporting emergency traffic, and security: Future directions’, Journal of Industrial Information Integration, Elsevier , 2021>
- English must be revised throughout the entire manuscript
- Limitations of the proposed algorithm are not highlighted
- Results section is weak, so authors are encouraged to add more results for the betterment of quality with clear insight
Author Response

(The authors gave the same response as above.)

Round 2
Reviewer 1 Report
.
Author Response
Nothing to add
Reviewer 3 Report
Authors do not seriosuly and properly adopted the reviewer's suggestions
Such as
- Comparison to the state of the art exisitng methods is missing
- Pseudocode of proposed method is missing
- Experiemtnal results are not sufficient and concinving
- Motivation, significance, novelty and solid contribution is not clearly written and hard to get convinced
So, I do not accept the paper in its current form
Author Response
Dear Reviewer 3
We are really sorry to hear about us not taking seriously the reviewers’ suggestions. It was not at all our intention, and we tried our best to answer all of them properly, given the tight deadline. 1 Comparison to the state of the art existing methods is missing As we pointed in the previous revision, this is a theoretical study of the importance of calibration for gaze estimation in low resolution environments using synthetic features, and, to the best of our knowledge, it is the first paper that addresses this issue. We would like to clarify that we are not proposing a novel method to solve gaze estimation in low resolution but studying the impact of the calibration process from a theoretical point of view. Although we could compare the obtained gaze estimation results with other studies, we are afraid that the comparison would not be fair and could be meaningless, as there are important differences, as for example: differences in the inputs (features vs images); differences in the nature of the data (synthetic vs real), differences in the network topology (FCN vs CNN)… However, thanks to your recommendation and the recommendations from the other reviewers, we have included and cited stated of the art techniques that yield conclusions like the ones extracted from our work. From our point of view, this is the most important comparison that can be made to state-of-the-art methods due to the theoretical nature of our work. 2. Pseudocode of proposed method is missing I would like to insist that in our paper it is not our purpose to propose a novel method for gaze-estimation. In our paper, the workflow of the experiments consists of, broadly speaking, a preprocessing step, a network training step, and a testing of the results. In none of these processes we are using an unusual method were the use of pseudoce would be necessary, from our point of view. We would like to highlight that the novelty from our work comes from the conclusions reached about the importance of the features used, features that can only be extracted from synthetic databases or by calibration and diverse computer vision tasks in real databases. However, as we proposed in the previous revision, an annex with flowchart and pseudocode for the experiment steps could be added if it could help to get a better comprehension of our work. 3. Experimental results are not sufficient and convincing. We are sorry about this, but please keep in mind the tight schedule we had to address your suggestions and the suggestions from the rest of the reviewers. In this time, we added two more elements to the results section, with their corresponding discussion. We personally thought that these results could improve the interpretability of our work, enhancing our conclusions. In such a brief deadline, we were clueless about what experiments we could perform to fill any gaps the reader could have when facing our work. 4. Motivation, significance, novelty and solid contribution is not clearly written and hard to get convinced As a result from your previous revision, we tried to make a clearer distinction of the points you emphasize. In the revised paper, we separated each of them in its own paragraph, to clarify each point. We would try to improve the read-flow to highlight them more.